# Ngn3-Positive Cells Arise from Pancreatic Duct Cells

**DOI:** 10.3390/ijms22168548

**Published:** 2021-08-09

**Authors:** Chiemi Kimura-Nakajima, Kousuke Sakaguchi, Yoshiko Hatano, Masahito Matsumoto, Yasushi Okazaki, Keisuke Tanaka, Takumi Yamane, Yuichi Oishi, Kenji Kamimoto, Ken Iwatsuki

**Affiliations:** 1Department of Nutritional Science and Food Safety, Faculty of Applied Bioscience, Tokyo University of Agriculture, Tokyo 156-8502, Japan; cnakajima@nig.ac.jp (C.K.-N.); sakaguchi.kousuke04@gmail.com (K.S.); yoshikoh41@gmail.com (Y.H.); ty204887@nodai.ac.jp (T.Y.); y3oishi@nodai.ac.jp (Y.O.); 2Laboratory of Mammalian Neural Circuits, National Institute of Genetics, Mishima 411-8540, Japan; 3Diagnostics and Therapeutics of Intractable Diseases, Intractable Disease Center, Advanced Diabetic Therapeutics & Department of Metabolic Endocrinology, Juntendo University, 2-1-1 Hongo, Bunkyo-ku, Tokyo 113-0033, Japan; mmatsumoto.bif@tmd.ac.jp (M.M.); ya-okazaki@juntendo.ac.jp (Y.O.); 4NODAI Genome Research Center, Tokyo University of Agriculture, Tokyo 156-8502, Japan; kt205453@nodai.ac.jp; 5Department of Developmental Biology, Washington University School of Medicine in St. Louis, 660 S. Euclid Ave., St. Louis, MO 63110, USA; kamimoto@wustl.edu

**Keywords:** Ngn3, pancreatic duct, endocrine cell, PDL, organoid culture, tuft cell

## Abstract

The production of pancreatic β cells is the most challenging step for curing diabetes using next-generation treatments. Adult pancreatic endocrine cells are thought to be maintained by the self-duplication of differentiated cells, and pancreatic endocrine neogenesis can only be observed when the tissue is severely damaged. Experimentally, this can be performed using a method named partial duct ligation (PDL). As the success rate of PDL surgery is low because of difficulties in identifying the pancreatic duct, we previously proposed a method for fluorescently labeling the duct in live animals. Using this method, we performed PDL on neurogenin3 (Ngn3)-GFP transgenic mice to determine the origin of endocrine precursor cells and evaluate their potential to differentiate into multiple cell types. Ngn3-activated cells, which were marked with GFP, appeared after PDL operation. Because some GFP-positive cells were aligned proximally to the duct, we hypothesized that Ngn3-positive cells arise from the pancreatic duct. Therefore, we next developed an in vitro pancreatic duct culture system using Ngn3-GFP mice and examined whether Ngn3-positive cells emerge from this duct. We observed GFP expressions in ductal organoid cultures. GFP expressions were correlated with Ngn3 expressions and endocrine cell lineage markers. Interestingly, tuft cell markers were also correlated with GFP expressions. Our results demonstrate that in adult mice, Ngn3-positive endocrine precursor cells arise from the pancreatic ducts both in vivo and in vitro experiments indicating that the pancreatic duct could be a potential donor for therapeutic use.

## 1. Introduction

The pancreas is an organ that serves two vital roles, exocrine and endocrine functions, which help digestion and regulate blood glucose, respectively. The exocrine part comprises acinar cells and a ductal system that secretes digestive enzymes and drains into the duodenum. The endocrine compartment of the pancreas consists of islets of Langerhans, which are composed of five types of endocrine cells: α cells, β cells, δ cells, ε cells, and PP cells. These cells secrete glucagon, insulin, somatostatin, ghrelin, and pancreatic polypeptide, respectively [1]. A deficiency in β cells results in diabetes mellitus, for which effective curative treatment has not been established. One promising therapy is transplantation of endocrine cells as a substitute for islet cells. However, the generation of β cells either from embryonic stem cells or induced pluripotent stem cells remains challenging because the culture conditions for terminal differentiation have not been well-established [2].

To find other sources of pancreatic stem cells with the potential to become functional β cells, researchers searched for adult stem cells in the adult pancreas [3]. However, the β-cell turnover rate is slow, and pancreatic regeneration can only be observed when the tissue is severely damaged [4]. In contrast, β cells dynamically repopulate in response to systemic increases in insulin during aging [5], pregnancy [6], obesity [7], and genetic insulin resistance [8], but the origin of these newly produced β cells is controversial. One group showed that pre-existing β cells are the major source of new β cells during adult life [4,9]. The other group suggested that β cells form in the adult by transdifferentiation of pancreatic acinar tissues [10]. Furthermore, another group showed that β cells can be generated from endogenous progenitors present in pancreatic ducts [11]. The mechanisms leading to replacement of the endogenous β-cell pool have been studied using chemical [12,13,14], genetic [15], and surgical methods such as partial pancreatomy [16] and partial duct ligation (PDL) [17].

PDL is a suitable model for mimicking human conditions that promote neoplasia; wide use of this model has provided novel insights into β cell formation. However, the outcomes of PDL appear to vary between laboratories, and some groups showed that PDL operation leads to a large increase in the β cell mass [11,15,17,18,19,20,21], whereas other groups failed to detect new β cells arising from the PDL pancreas [22,23,24]. The discrepancy in these results is attributed to the success rate of the operation, as performing PDL is difficult. To overcome this problem, we established a new technique for PDL using a simple imaging approach [25]. We examined whether endocrine progenitor cells emerge from the pancreatic duct after PDL by introducing neurogenin3 (Ngn3)-GFP mice [26], as Ngn3 is a well-established marker of endocrine progenitor cells [27]. By tracing the GFP expression after a PDL treatment, we hypothesized that regenerating ductal cells could give rise to endocrine cell lineages. To examine this hypothesis, we introduced a ductal organoid culture system to examine whether Ngn3-positive cells emerge accompanied by endocrine markers as described by Huch et al. in 2013 [19]. To the best of our knowledge, this is the first study to demonstrate that pancreatic-ductal-derived cells can differentiate into endocrine lineages.

## 2. Results

### 2.1. Observation of Ngn3-Positive Cells in the Ligated Part of the Pancreas at 3, 5, 6, and 21 Days after Surgery

First, we performed PDL at the tail region of the pancreas in Ngn3-GFP mice. Ngn3-positive endocrine precursor cells, marked by GFP, were first observed in the ligated pancreas 3 days after PDL surgery (day 3, Figure 1B). GFP signals were also found in the tissues on days 5, 6, and 21 (Figure 1C–F), although we did not detect any GFP signals in the sham-operated pancreas (Figure 1A). GFP signals were observed outside of the lining of CK19-positive pancreatic ductal cells (Figure 1B–E). Moreover, we observed some GFP-positive cells in CK19-positive pancreatic ductal cells (Figure 1D, arrows). To examine whether Ngn3-positive cells express a definitive marker for islet hormones, we performed immunohistochemistry analysis but did not detect hormones such as insulin, glucagon, or somatostatin in Ngn3-positive cells on day 6 (data not shown).

### 2.2. Generation of Pancreatic Ductal Organoids from Ngn3-GFP Mice

Because we observed temporal expansion of Ngn3-GFP-positive cells after PDL from the proximal region of CK19-positive ducts, we hypothesized that GFP-positive cells arose from the pancreatic ducts. To clarify the emergence of endocrine progenitor cells from pancreatic ducts, we generated pancreatic ductal organoids from Ngn3-GFP transgenic mice. We first observed cell proliferation around the edge of the duct on day 2 of culture, and continuous expansion of the organoid was observed (Figure 2A–D). On day 4, we performed immunostaining of the organoids with CK19 antibodies and found that most organoids expressed CK19 (Figure 2E,F). In contrast, amylase, a marker of acinar cells, was not observed (Figure 2G), nor were insulin, glucagon, or somatostatin (Appendix A Appendix A).

Next, we examined whether Ngn3-GFP-positive cells were present in the ductal organoids. We detected no GFP signal before day 5 in the organoid cultures (Figure 3A). GFP signals were first observed in the organoids at around days 7–9 (Appendix A Appendix A, Figure 3B). The number of GFP-positive cells increased within the same organoid up to day 11 (Figure 3B–D).

### 2.3. Ngn3-Positive Cells Can Differentiate into Endocrine Cells

To clarify the properties of Ngn3-expressing organoids, we extracted total RNA on day 11 of organoid culture as described in the Materials and Methods. RNA-Seq was performed to compare gene expression between organoids that expressed GFP signals and those that did not (Figure 4A). We confirmed that organoids showing a GFP signal predominantly expressed higher mRNA levels of Ngn3, demonstrating the reliability of the data (Figure 4B). The expression of transcription factors such as NeuroD1 and Isl1 was elevated in Ngn3-expressing organoids compared to organoids that did not express Ngn3 (Figure 4C,D). We observed no significant increase in the gene expression of glucagon (Gcg) but detected Gcg expression in both the Ngn3(-) and Ngn3(+) fractions (Figure 4E). The expression of endocrine cell markers, such as Sst, Ghrl, and Chga, was significantly elevated in the Ngn3(+) fraction (Figure 4F–H, *q* < 0.05). Interestingly, the expression of molecules involved in taste information transmission, such as Trpm5, Gnat3, and Pou2f3, was also significantly elevated (Figure 4I–K, *q* < 0.05). To confirm the RNA-seq results, we performed RT-PCR with specific primers for each analyzed gene (Figure 4L). These results showed that pancreatic-duct-derived cells can differentiate into both the enteroendocrine and tuft cell lineages.

## 3. Discussion

Ngn3 is a transcription factor essential for endocrine differentiation and often used as a marker to show endocrine lineage specification [27]. We demonstrated through both in vivo and in vitro experiments that Ngn3-positive endocrine precursor cells emerge from the pancreatic duct. Ngn3-GFP mice do not express GFP in the normal adult pancreas (Figure 1A, Appendix A Appendix A); however, we showed that GFP-positive cells emerge after PDL treatment in vivo or in organoid culture using dissected pancreatic ducts. Immunohistochemistry (Figure 2, Appendix A Appendix A) and transcriptome analysis (data not shown) suggested that pancreatic organoids in which most of the cells showed immunoreactivity to CK19 do not initially express any components derived from mature acinar or endocrine cells. However, after 3–5 days in vivo or 7–9 days in organoid culture, we observed the emergence of GFP-positive cells, indicating Ngn3 was expressed. We further demonstrated by transcriptome analysis that pancreatic-duct-derived Ngn3-positive cells have the potential to differentiate into endocrine lineages. Our results are consistent with the report that Ngn3-expressing cells derived from human pluripotent stem cells are endocrine progenitor cells and can be further differentiated to produce insulin, glucagon, and somatostatin [28]. Therefore, the molecular mechanism underlying endocrine cell differentiation of the pancreas appears to be similar between human and mouse, and thus can be extrapolated from a mouse model to human use for providing donor cells for regeneration therapies.

Through our data, we showed that pancreatic duct cells have the potential to dedifferentiate to become Ngn3-expressing endocrine progenitor cells (Figure 5), although it should be further examined whether the phenomenon that we observed was through stem cells or direct conversion from duct cells to Ngn3-positive cells. There have been several hypotheses regarding the existence and location of adult stem- or progenitor-like cells that can give rise to functional β cells. Cell ablation methods using chemical [12,13,14], genetic [15], and surgical techniques [16,17] have been used to test the regeneration activity of the adult mouse pancreas, which is capable of yielding new insulin-producing β cells. Although the PDL technique is widely used to induce pancreatic regeneration, in β-cell neogenesis followed by PDL, the origin of β cells remains controversial. Our newly established PDL method that can easily visualize the pancreatic duct helped us to avoid simultaneous ligation of blood vessels or nerve fibers, thus increased the reliability of PDL operation [25,29]. PDL on Ngn3-GFP transgenic mice using this method revealed that GFP-positive cells often emerged proximal to the reorganizing ducts. Therefore, we hypothesized that Ngn3-positive cells arise, at least in part, from the pancreatic duct when the pancreas is damaged. As most Ngn3-positive cells did not overlap with the pancreatic duct, we predicted that Ngn3-positive cells can only convert into endocrine cells after they move out of the pancreatic duct. This idea was previously suggested by Shih et al., who showed that ductal markers and Ngn3 expression are almost exclusive [30]. Further studies are needed to determine whether Ngn3-positive cells arise from pancreatic stem cells or from the dedifferentiation of functional cells in the pancreas. Several reports support the idea of acinar-to-ductal transdifferentiation, proposing that acinar cells also have plasticity [31,32,33,34]. Our data, both in vivo and in vitro, show that Ngn3-positive cells can be produced from pancreatic ductal cells; however, our results do not exclude the possibility of acinar-to-ductal transdifferentiation.

Notably, we found that the expression of molecules specific to tuft cells such as Trpm5, Gnat3, and Pou2f3 [35], was significantly upregulated in correlation with the expression of Ngn3. These results suggest that pancreatic ducts can differentiate into multiple cell lineages including tuft and endocrine cells. Both cell types emerged from the same Ngn3-GFP-positive cell fractions, indicating that cell differentiation occurs simultaneously, although lineage tracing experiments are needed to confirm this. It is natural to observe tuft cell differentiation in organoid culture because these cells were reported to be present in the pancreatic duct in mice and humans [35,36]. Recent studies showed that tuft cells trigger type 2 immunity after parasite infection or when stimulated by succinate [37,38,39,40]. Based on previous studies, we predicted that pancreatic ducts also possess a self-defense function, likely to protect the tissue from microorganisms and irritants confronted by ductal tuft cells when the duodenal reflux or pancreatitis occurs. In our future studies, we will establish a culture method to obtain fully mature endocrine or tuft cells that can respond to nutrients or hazardous substances. Our in vitro culture methods utilizing Ngn3-GFP transgenic mice will help to optimize the conditions for cell differentiation, as cell maturation appears to occur simultaneously. There is, however, a limitation to this study: organoids generated from pancreatic duct might not always recapitulate in vivo to produce Ngn3-expressing progenitor cells as we expected. It is important to find the appropriate culture condition for organoids to differentiate into certain lineages.

In conclusion, we showed both in vivo and in vitro that the pancreatic duct can produce endocrine progenitor cells that express Ngn3. The organoid culture generated from the pancreatic duct will be a good model to study endocrine cell differentiation as well as to understand how the pancreas is regenerated after injury. In the future, the pancreatic duct organoid will potentially be a new source for endocrine cells for therapeutic usage as well as in vitro drug screening system to create new drugs for pancreatic cell regeneration.

## 4. Materials and Methods

### 4.1. Animals

All animals used in this study were obtained, housed, cared for, and used in accordance with the “Guiding Principles in the Care and Use of Animals” published by the Animal Care Committee of Tokyo University of Agriculture (Tokyo, Japan, Ethics identification number: 290084, 28 September 2017). Ngn3-GFP mice were described previously [26]. Mice bred on a C57BL/6 background and crossed with C57BL/6 or ICR mice were maintained on a 12 h light cycle with access to food and water ad libitum.

### 4.2. PDL after Cholyl-Lysyl-Fluorescein Injection

A combination anesthetic was prepared from medetomidine (0.3 mg/kg), midazolam (4.0 mg/kg), and butorphanol (5.0 mg/kg). Medetomidine was purchased from Zenoaq (Tokyo, Japan). Midazolam was purchased from Sandoz (Tokyo, Japan). Butorphanol was purchased from Meiji Seika Pharma (Tokyo, Japan). Chlorhexidine gluconate was purchased from Sumitomo Dainippon Pharma (Osaka, Japan). Cholyl-lysyl-fluorescein was purchased from Corning (Corning, NY, USA). PDL surgeries were performed as previously described [25].

### 4.3. Immunohistochemistry

The following antibodies were used: anti-pancreatic alpha amylase (ab21156, Abcam, Cambridge, U.K.), anti-insulin (sc-7839, Santa Cruz Biotechnology, Dallas, TX, USA), anti-somatostatin 28 (ab22682, Abcam), anti-glucagon (ab92517, Abcam), and anti-GFP (Rockland Immunochemicals, Gilbertsville, PA, USA). Rabbit anti-CK19 antibodies were a gift from Dr. Atsushi Miyajima (The university of Tokyo) [41,42]. PDL-administered pancreas or pancreatic organoids from Ngn3-GFP mice (n = 3, respectively) were fixed in 4% paraformaldehyde in phosphate-buffered saline and embedded in O.C.T. compound (Sakura Finetek, Torrance, CA, USA). Frozen sections (12 µm) were prepared and incubated in 0.3% Triton X-100 and 2% donkey serum for 1 h, followed by incubation with the primary antibodies overnight at 4 °C. Next, the sections were incubated with secondary antibodies (Alexa Fluor 555 or 594 IgG; Invitrogen, Carlsbad, CA, USA) and counterstained with DAPI (Nacalai Tesque, Kyoto, Japan).

### 4.4. Organoid Culture

Ngn3-GFP mouse pancreatic organoids were prepared as previously described [19,43] with minor modifications. Briefly, the pancreas was dissected and minced into pieces, followed by incubation with digestion solution (wash medium with 0.125 mg/mL collagenase IV, 0.125 mg/mL dispaseⅡ, and 0.1 mg/mL DNase I) for 1 h at 37 °C. Isolated ducts were embedded in Matrigel covered with the culture medium (5% R-spondin2-conditioned medium, 5% Noggin-conditioned medium, 2% B27 supplement (without vitamin A), 100 ng/mL recombinant human FGF10 (Peprotech, Rocky Hill, NJ, USA), 50 ng/mL recombinant mouse epidermal growth factor (Peprotech), 10 mM nicotinamide (FUJIFILM WAKO, Osaka, Japan), 1 mM N-acetylcysteine (Sigma Aldrich, St. Louis, MO, USA), and 10 nM recombinant human [Leu15]-gastrin1 (Sigma Aldrich) in the basal medium). Every 2–3 days, the medium was replaced with fresh medium.

### 4.5. RNA-Seq Analysis

Total RNA was extracted from the organoids using ISOGEN (Nippon Gene, Tokyo, Japan). The quality and quantity of the extracted RNA were determined using a spectrophotometer (Nanodrop 2000, Thermo Fisher Scientific, Waltham, MA, USA) and an Agilent 2100 Bioanalyzer (Agilent Technologies, Santa Clara, CA, USA). Library preparation was performed using a NEBNext Ultra RNA Library Prep Kit for Illumina (New England Biolabs, Ipswich, MA, USA) according to the manufacturer’s protocol. The precise concentration of the libraries was determined using a KAPA Library Quantification Kit (Kapa Biosystems, Wilmington, MA, USA). All libraries were diluted to a concentration of 10 nM and mixed in equal amounts. The library mixture was sequenced by 1 × 100 base pair single-read sequencing on an Illumina HiSeq 2500 (Illumina, San Diego, CA, USA). Reads in FASTQ format were generated using bcl2fastq2 conversion software (Illumina, version 2.18). The read data were submitted to the DDBJ Read Archive (accession number: DRA007996). Adapter sequences and the initial 13 bases in each read were removed using CLC Genomics Workbench 10 software (Qiagen, Hilden, Germany). The cleaned read data were mapped to the mouse reference genome (GRCm38.p6) retrieved from the Ensembl genome browser (https://www.ensembl.org/index.html, accessed on 22 November 2018). Mapping parameters were as follows: mismatch cost, 2; insertion cost, 3; deletion cost, 3; length fraction, 0.8; and similarity fraction, 0.8.

### 4.6. Statistical Analysis

Differentially expressed genes of Ngn3-GFP-expressing organoids and control organoids were identified with significant thresholds of a fold-change ≥ |1.5| and false discovery rate adjusted *p*-value (*q*-value) < 0.05 using a generalized linear model approach using the CLC Genomics Workbench built-in tools Differential Expression for RNA-Seq.

### 4.7. Reverse Transcription (RT)-PCR

Organoids were collected after 11 days, and total RNA was extracted using ISOGEN. The SuperScript III First-Strand Synthesis System for RT-PCR (Thermo Fisher Scientific) was used to generate cDNA. RT-PCR was performed using primer pairs against the gene of interest (primers are shown in Appendix A). RT-PCR was performed as previously described [44] using primers specific for enteroendocrine and tuft cell markers (Appendix A Appendix A).

## Figures and Tables

**Figure 1 ijms-22-08548-f001:**
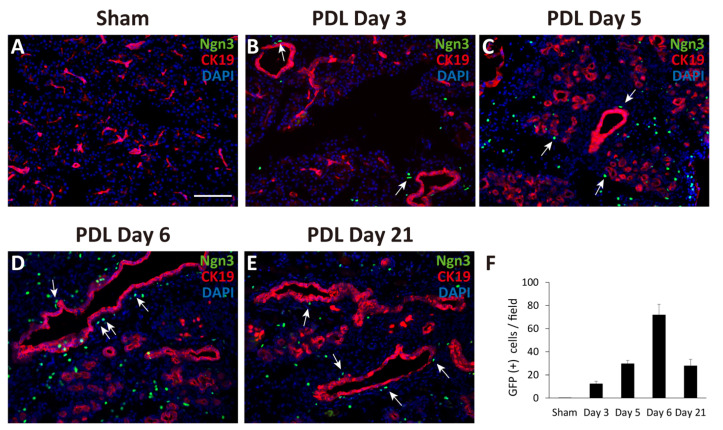
Monitoring GFP expression in pancreatic sections after partial duct ligation (PDL) using Ngn3-GFP transgenic mice. The tail region of the pancreas was ligated and examined for GFP expression using anti-GFP antibodies (green). Pancreatic ducts were identified using anti-CK19 antibodies (red). GFP expression was not detected in the sham-operated tail region of the pancreas (**A**). GFP signals appeared at around day 3 and gradually increased until day 6 after PDL surgery (**B**–**E**). The average number of GFP-positive cells in one micrograph is shown as a bar graph (**F**, n = 3). Ngn3-positive cells were often detected proximal to CK19-positive ducts after PDL surgery (arrows) (bar: 100 µm).

**Figure 2 ijms-22-08548-f002:**
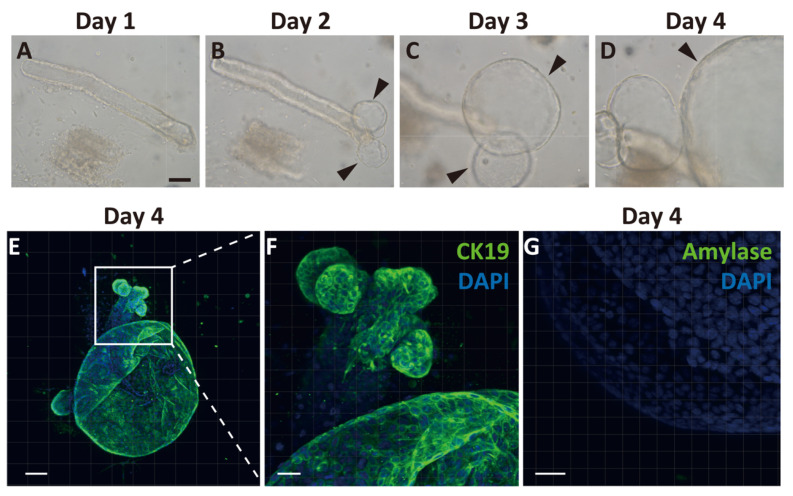
Generation and characterization of ductal organoid from the pancreas. Pancreatic ducts were dissected from the pancreas and embedded in Matrigel covered with organoid media to start organoid culture (**A**). After 2 days, the sheared end of the duct started proliferating and then formed organoids (**B**–**D**, arrowheads). Immunohistochemical analysis of organoids after 4 days of culture. All cells within the organoids expressed cytokeratin 19 (CK19), a selective ductal marker (**E**,**F**). An acinar cell-specific marker, amylase, was not detected on day 4 (**G**). (bars: **A**–**D**, 500 µm; **E**, 100 µm; **F**, 50 µm; **G**, 30 µm).

**Figure 3 ijms-22-08548-f003:**
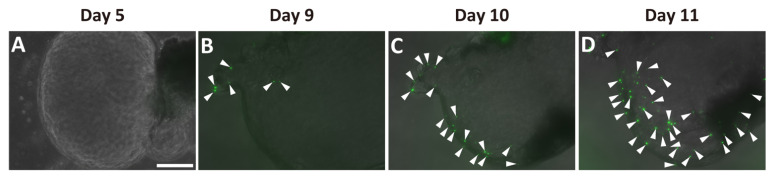
Emergence of Ngn3-GFP-positive cells in pancreatic ductal organoid. Ductal organoids were generated from Ngn3-GFP mice, and GFP expression, representing Ngn3 expression, was monitored. Ngn3-GFP expression was not observed on day 5 (**A**), but first appeared around days 7–9 of culture (**B**). GFP-positive cells started increasing thereafter (**B**–**D**, arrowheads) (bar: 250 µm).

**Figure 4 ijms-22-08548-f004:**
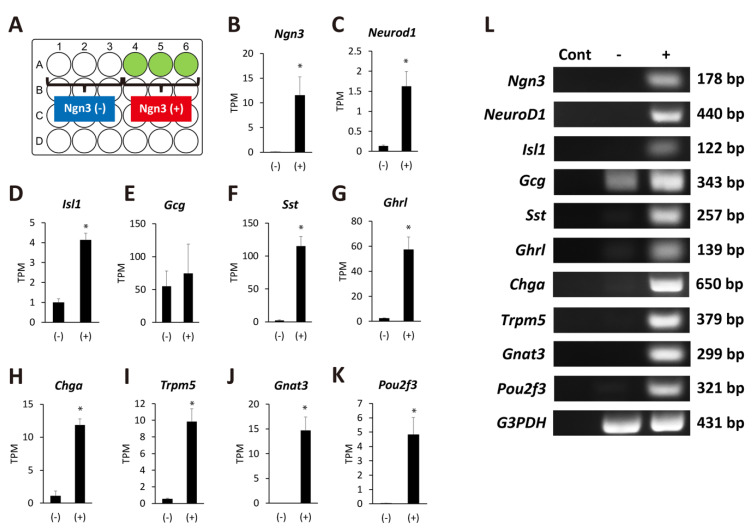
Ngn3-GFP-positive organoids show the potential to differentiate into several lineages. (**A**) Schematic presentation of the wells with or without organoids expressing GFP. Organoids were defined as Ngn3(+) if GFP-positive cells were observed on day 11. In contrast, organoids without GFP signal were defined as Ngn3(−). (**B**–**K**) Transcriptome analysis was performed with total RNAs extracted from day 11 organoids that expressed either Ngn3(−) and Ngn3(+) cells. Ngn3, NeuroD, and Isl1, key regulators of the endocrine lineage; mature endocrine markers Sst, Ghrl, and Ghga; and taste-related genes Trpm5, Gnat3, and Pou2f3, were significantly expressed in Ngn3-GFP-expressing organoids (n = 3, * *q* < 0.05). The gene Gcg, encoding GLP-1 and GLP-2, was expressed in both Ngn3(−) and Ngn3(+) cells (**E**). Results of RNA-Seq analysis were confirmed by RT-PCR (**L**). Cont, negative control without cDNA.

**Figure 5 ijms-22-08548-f005:**
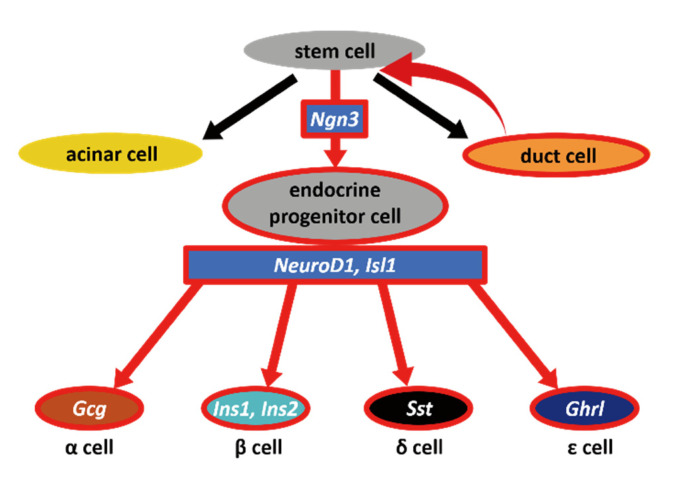
Schematic diagram of our hypothesis. In vivo and in vitro results indicated that dedifferentiation occurs in the pancreatic duct when ducts are severely injured. As a result, Ngn3-positive cells emerge from pancreatic stem cells; these stem cells have the potential to become mature endocrine cells, acinar cells, as well as duct cells.

## Data Availability

All the data are shown in the main manuscript and in the Appendix A.

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
