# Peer review of "Ngn3-Positive Cells Arise from Pancreatic Duct Cells"

_ijms, 2021, doi:10.3390/ijms22168548_

Round 1
Reviewer 1 Report
This is a very interesting study that analyzed whether endocrine progenitor cells emerge from the pancreatic duct after partial duct ligation in mice. The study is sound however certain issues have to be addressed before publication.
Please see attached PDF for a point-by-point commentary.
cultured in vitro could be a good 209 source of both endocrine cells as well as other cell types such as tuft cells. In the future, 210 the pancreatic duct organoid would be a new source for endocrine cells for therapeutic 211 usage as well as in vitro drug screening system

Author Response
Response to the Reviewer 1
We appreciate the Reviewer 1 for the constructive comments on our manuscript.
Reference: As reviewer 1 has suggested, we have added one reference within Introduction that describe endocrine cells within pancreas.
Title:
Thank you for your suggestion to change the title of our manuscript. We agree that our findings will provide us with a novel therapeutic tool. However, we hesitate to include the idea in the title because we are using a rodent model so that we should move onto examine with primate tissue. After the primate study, we could emphasize on therapeutic use of duct cells. Instead of changing the title we describe in the manuscript that our work could contribute to the human therapies in the future. Please see our discussion referring to regeneration therapies and therapeutic usage of pancreatic duct cells (line 170-173 and 219-225).
Protocols:
We appreciate your comments on the protocol. According to your suggestion, we try to guide the readers to the Methods where we precisely described how total RNA extraction and RNA-seq analysis were carried out (line 268-285).
We also appreciate the reviewer 1 for correcting our typos.
Thank you very much for helping to improve our manuscript.
Reviewer 2 Report
The study by Kimura-Nakajma and co-authors represents, in my opinion, a significant advance in the quest towards endocrine cell therapy. The results are encouraging (at least in the context of the mouse system) and the manuscript is well-written. A minor comment: please make it clear that you are referring to the tail region of the pancreas, and not the tail region of the mouse (line 78 and elsewhere). It would also add to the discussion section, in my opinion, if the possibilities of extending the observed results to humans could be addressed.
Author Response
Response to the Reviewer 2
We are most grateful to the reviewer for the helpful comments on our manuscript.
・As suggested by the reviewer, we have changed the sentence about “tail region of the mouse pancreas” but not tail region of the mouse (line 81 in the revise manuscript).
・To answer the comment of the reviewer, we have rewritten the Discussion part and included future possibilities for human regeneration therapies (line 166-172). Accordingly, we added one reference #28 describing human pluripotent stem cells.
All the changes we made are marked in Green including changes made to reply for other Reviewer.
We thank Reviewer 2 for comments to improve our manuscript.
Reviewer 3 Report
The research work carried out by the authors is very interesting and innovative. The conclusions are missing. The introduction and discussion should be expanded.
Author Response
Response to the Reviewer 3
We thank to the Reviewer 3 for the helpful comments and have changed our manuscript as has been suggested.
The comments by reviewer have led to a stronger manuscript.
All the changes we made are marked in Green including changes made to reply to other Reviewers.
Conclusions:
・According to the reviewers comment, we have rewritten the Conclusion for the readers to clearly understand the data we have presented (Line 219-222).
Expansion of Introduction and Discussion:
We have expanded Introduction and Discussion as reviewer 3 has suggested.
・We added/changed sentences in the introduction to raise our hypothesis and make our aim clear (Line 71-75).
・In Discussion, we added/changed sentences to show that our mouse model could extrapolate to human study because the mechanism of endocrine differentiation seems to be similar between mouse and human (Line 167-171). We also changed the last paragraph of discussion to focus more onto our conclusion (Line 219-225).